# ALV-miRNA-p19-01 Promotes Viral Replication via Targeting Dual Specificity Phosphatase 6

**DOI:** 10.3390/v14040805

**Published:** 2022-04-13

**Authors:** Yiming Yan, Sheng Chen, Liqin Liao, Shuang Gao, Yanling Pang, Xinheng Zhang, Huanmin Zhang, Qingmei Xie

**Affiliations:** 1Guangdong Laboratory of Lingnan Modern Agriculture, College of Animal Science, South China Agricultural University, Guangzhou 510642, China; yimingyan@stu.scau.edu.cn (Y.Y.); chens@stu.scau.edu.cn (S.C.); lapchin_l@outlook.com (L.L.); 13600478231@163.com (Y.P.); xinhengzhang@yeah.net (X.Z.); 2Guangdong Haid Institute of Animal Husbandry & Veterinary, Guangzhou 511400, China; sunnie_scau@163.com; 3Avian Disease and Oncology Laboratory, US National Poultry Research Center, Agriculture Research Service, USDA, East Lansing, MI 48823, USA; jqyjs19940723@163.com

**Keywords:** ALV-miRNA-p19-01, ALV-J, DUSP6, ERK2

## Abstract

MicroRNAs (miRNAs) are a group of regulatory noncoding RNAs, serving as major regulators with a sequence-specific manner in multifarious biological processes. Although a series of viral families have been proved to encode miRNAs, few reports were available regarding the function of ALV-J-encoded miRNA. Here, we reported a novel miRNA (designated ALV-miRNA-p19-01) in ALV-J-infected DF-1 cells. We found that ALV-miRNA-p19-01 is encoded by the genome of the ALV-J SCAU1903 strain (located at nucleotides site 779 to 801) in a classic miRNA biogenesis manner. The transfection of DF-1 cells with ALV-miRNA-p19-01 enhanced ALV-J replication, while the blockage of ALV-miRNA-p19-01 suppressed ALV-J replication. Furthermore, our data showed that ALV-miRNA-p19-01 promotes ALV-J replication by directly targeting the cellular gene dual specificity phosphatase 6 through regulating ERK2 activity.

## 1. Introduction

Retroviruses are a large family of enveloped RNA viruses, involving a variety of diseases in a wide range of host species [1]. The genome of retroviruses usually integrates with the host genome, producing viral proteins and genomic RNA to assemble the progeny viral particles [2]. Retroviruses are divided into seven groups: five of these groups possess oncogenicity, which are known as oncoviruses, and the other groups are lentiviruses and spumaviruses [3]. The oncoviruses, including the Rous sarcoma virus (RSV), avian leukosis virus (ALV) and mouse sarcoma virus (MSV), are the largest subfamily of the retrovirus family, and are implicated in a series of human and animal tumor diseases, such as leukemias, mammary tumors and skin cancer [4,5,6].

MicroRNAs (miRNAs) are a group of small non-coding RNAs of 20–24 nucleotides (nt) in length, serving as major regulators with a sequence-specific manner in multiple biological processes [7,8]. A series of viral families have been proven to encode miRNAs [9]. The first viral-encoded miRNA was found in Epstein-Barr virus (EBV)-infected cell lines in 2004 [10]. To date, more than 530 virus-encoded miRNAs have been identified, and over 95% of these miRNAs were encoded by herpesvirus. It has been reported that the HIV genome encodes several miRNAs, such as miRNA-TAR and miRNA-H1 [11,12]. Interestingly, a recent report indicated that a small RNA (designated E (XSR) miRNA) was encoded within the E(XSR) element of ALV-J [13].

Viral release is the critical step in the life cycle of retroviruses, with an important role in the formation of viral particles. Retroviral release usually takes place at the plasma membrane. There is increasing evidence supporting the idea that the gag protein of retroviruses hijacks the multivesicular body (MVB) machinery to promote viral release [14,15,16]. The gag protein binding to the membrane is an essential step in viral replication, and viral protein p19 is deemed to drive gag polyprotein to the plasma membrane and induce viral release [17,18,19]. In this study, we identified a novel miRNA (designated ALV-miRNA-p19-01) encoded by the p19 gene of the ALV-J SCAU1903 strain. The epigenetic upregulation of ALV-miRNA-p19-01 promoted viral replication in vitro. Moreover, the positive effect of ALV-miRNA-p19-01 on viral replication is associated with its binding to dual specificity phosphatase 6 (DUSP6), which regulates the ERK2 activity.

## 2. Materials and Methods

### 2.1. Ethics Statement

The animal study protocol was approved by the Committee of Animal Experiments of South China Agricultural University (approval ID: SYXK-2019-0136). All study procedures and animal care activities were conducted per the recommendations in the Guide for the Care and Use of Laboratory Animals of the National Institutes of Health.

### 2.2. Virus, Cell Lines and Animals

DF-1 cells were cultured in Dulbecco’s Modified Eagle Medium (DMEM) supplement with 10% fetal bovine serum (FBS). The primary chicken embryo fibroblast cells (CEF) were extracted from 9-day-old specific pathogen-free (SPF) chicken embryos. One-day-old SPF cocks were purchased from Guangdong Wen′s Foodstuffs Group Co Ltd. (Yunfu, China). The ALV-J SCAU1903 strain (No. MT175600) isolated from a commercial layer flock was obtained in our lab.

### 2.3. Chemicals and Antibodies

The anti-actin, anti-Dicer, anti-Drosha, anti-ERK and anti-p-ERK antibodies were from CST (Boston, MA, USA). The anti-p27 antibodies were purchased from Qianxun Biological (Guangzhou, China). Anti-rabbit IgG and -mouse IgG antibodies were from Proteintech (Chicago, IL, USA). The ERK2 inhibitor MK-8353 was purchased from Selleck (Shanghai, China). The α-amanitin was purchased through Sigma–Aldrich (St. Louis, MO, USA).

### 2.4. Identification of ALV-J-Positive Chickens

A total of 20 one-day-old SPF cocks were randomly divided into two groups (ten birds per group), and independently maintained in negatively pressured biosecurity isolators under quarantine conditions. The birds in Group I were inoculated with a dose of 103.5 TCID50 ALV-J SCAU1903 strain, while the birds in Group II were inoculated with the same volume of nutrient solution. Double inoculation was performed at a 5-day interval. The infectious status in each bird was determined every 2 weeks using reverse transcription PCR (RT-PCR), virus isolation and ELISA assay as described previously [20]. The experiment period lasted 20 weeks.

### 2.5. Transmission Electron Microscopy

DF-1 cells were transfected with ALV-miRNA-p19-01 or miR-NC, si-DUSP6 or si-NC, and then infected with rSCAU1903-mut. The cells were fixed with 2.5% glutaraldehyde for 2 h, washed with PBS, and fixed with 1% OsO_4_ solution for 4 h. The cells were dehydrated in a series of ethanol of increasing concentration (30%, 50%, 70%, 95% and 100%), and embedded in Epon–Durcupan resin. Polymerized blocks were cut into 100 nm thin sections, collected on 200 mesh size copper grids with square or hexagonal openings, and stained with a saturated aqueous solution of uranyl acetate for 4 min. The sections were examined using Mega View III CCD camera.

### 2.6. RNA Library Construction and miRNA Sequencing

Small RNA was extracted from DF-1 cells using the mirVana miRNA isolation kit (Ambion, Austin, TX, USA) according to the manufacturer’s protocol. The samples were qualified and sent to Beijing Genomics Institute Shenzhen Co. (BGI-Shenzhen) for sequencing using Solexa technology. The reads were aligned with both the gallus genomic sequence and the genomic sequence of ALV-J SCAU1903. The secondary-structure analysis and prediction of the precursor were performed using miRDeep2 software.

### 2.7. RT-PCR

Total RNA was extracted from cells or organs using TRIzol reagent (TaKaRA, T9108) and subjected to a reverse transcription reaction for cDNA synthesis. QRT-PCR was used to detect the transcriptional levels of target genes and miRNAs using Bimake SYBR Green qPCR Master Mix and TaqMan^®^ miRNA assays (Thermo Fisher Scientific, Carlsbad, CA, USA), respectively. β-actin and U6 RNA were used as the reference controls. The relative mRNA and miRNA expression levels were calculated by the comparative threshold cycle method (2^−ΔΔCt^). The primers used in this study are listed in Appendix A.

### 2.8. Northern Blotting Analysis

A total of 10 μg of total RNA was loaded on a 15% denaturing polyacrylamide gel, and then transferred onto a nylon membrane (GE Healthcare, Waukesha, WI, USA). Biotin-labeled oligonucleotide probes complementary to ALV-miRNA-p19-01, were used for hybridization. The Chemiluminescent Nucleic Acid Detection Module (Thermo Scientific) was used for the detection of biotin luminescence according to the manufacturer’s instructions. The chicken U6 RNA was used as the reference control.

### 2.9. TCID_50_ Assay

DF-1 cells were transfected with ALV-miRNA-p19-01 or miR-NC, ASO-ALV-miRNA-p19-01 or ASO-NC, pRK5-Flag-DUSP6 or pRK5-Flag, si-DUSP6 or si-NC, or treated with MK-8353 or DMSO. At 24 h post-transfection or 12 h post-treatment, cells were infected with ALV-J SCAU1903 or ALV-J rSCAU1903-mut strain at a dose of 103.0 TCID_50_, or DMEM were used as controls. At 48 or 72 h post-infection (hpi), the cellular supernatants were harvested to examine p27 antigen by an avian leukosis virus antigen test kit (IDEXX, Inc., Westbrook, MA). The TCID_50_ values in the supernatant were calculated according to the Reed–Muench method.

### 2.10. Western Blotting Analysis

The cells and cellular debris were removed from the harvested virions-containing supernatant with a low-speed spin. Then, virions were centrifuged through a 20% sucrose cushion at 100,000× *g* for 2 h at 4 °C. The resulting pellet containing virions was suspended in PBS buffer. Cells or virions were lysed using lysis buffer containing protease inhibitor and centrifuged at 10,000× *g* for 10 min for sample collection. Samples were fractionated by SDS-PAGE, transferred to polyvinylidene fluoride membrane and blocked with 5% skimmed milk. After incubation with appropriate primary and secondary antibodies, blots were developed using an ECL kit (Beyotime, Shanghai, China).

### 2.11. RNA Pull-Down Assay

The probes which included biotinylated ALV-miRNA-p19-01 WT and control random RNA were incubated with DF-1 cell lysates for 1 h at 4 °C. The RNA–RNA complexes were captured by streptavidin magnetic beads. Finally, the beads were washed by PBS and extracted miRNA–mRNA complexes. The differential mRNA expression was detected via RNA-seq [21].

### 2.12. Luciferase Reporter Vector Construction

A fragment of the 3ʹ UTR of wild-type DUSP6 was amplified by RT-PCR using the primers 5ʹ-GGAAGGCGACGAGCTCTGGG-3ʹ and 5ʹ-GGTGTCTAGACCACAGGCAGCAGGAGAC-3ʹ. The region containing the putative binding sites for ALV-miRNA-p19-01 was inserted downstream of the stop codon of firefly luciferase in the pmiRGLO Dual-Luciferase miRNA Target Expression Vector (Promega, Madison, WI, USA) (designated DUSP6-3ʹUTR-wt). PmiRGLO-DUSP6-3ʹUTR-mut (designated DUSP6-3ʹUTR-mut), which carries a mutated version of the complementary site for the seed region of ALV-miRNA-p19-01, was generated using the primers 5′-GGAAGGCGACGAGCTCTGGG-3′ and 5′-GGAAGGCGACGAGCTCTGGG-3′.

### 2.13. Construction of Infectious Proviral Clone of ALV-J

The DNA fragment of the whole proviral genome of SCAU1903 and the linearized pVAX1 vector were processed by homologous recombination to rescue the recombinant virus (designated pVAX-ALV-SCAU-wt) (Appendix A). In brief, three fragments of proviral cDNA of the SCAU1903 strain and the linearized pVAX1 vector were generated by polymerase chain reaction (PCR), and then fused using the Seamless Assembly Cloning Kit (Clone Smarter, USA). DF-1 cells were transfected with proviral DNA and the virus was rescued. The ALV-miRNA-p19-01 precursor sequence was mutated in the pVAX-ALV-SCAU-wt to rescue the mutated recombinant virus (designated pVAX-ALV-SCAU-mut) (Appendix A).

### 2.14. Statistical Analysis

Data were processed using GraphPad Prism (version 5.0) and expressed as the mean ± SE. The Student’s *t*-test was used to assess differences among groups; *p* < 0.05 and *p* < 0.01 were considered to show significant differences between groups.

## 3. Results

### 3.1. Identification of ALV-J-Encoded miRNAs by Deep Sequencing

To identify the miRNA component of ALV-J-infected DF-1 cells and normal DF-1 cells, we isolated total RNA from ALV-J-infected DF-1 cells and normal DF-1 cells, and employed a deep sequencing assay to profile small RNA components. As a result, we identified a novel RNA transcript of approximately 23 nt in length, which was only found in ALV-J-infected DF-1 cells, but not in normal DF-1 cells (Figure 1A). Sequence analysis showed that the novel RNA transcript (designated ALV-miRNA-p19-01) is homologous with the ALV-J SCAU1903 transcript, which was located in the gag gene of the ALV-J SCAU1903 (Figure 1B). Considering the association of miRNA formation with the precursor structure, we analyzed the secondary structure of the precursor sequence of ALV-miRNA-p19-01 using the miRDeep2 software. As a result, the precursor sequence of ALV-miRNA-p19-01 formed a classic stem-loop structure of miRNA, and ALV-miRNA-p19-01 was located in the 3′-end of the predicted precursor (Figure 1C). Furthermore, we aligned the sequence of ALV-miRNA-p19-01 with reference sequences retrieved from the GenBank database and found that the sequence of ALV-miRNA-p19-01 existed in the gag gene among different ALV strains (Figure 1D and Appendix A).

### 3.2. Experimental Validation of ALV-miRNA-p19-01

To confirm the presence of ALV-miRNA-p19-01, we employed RT-PCR and Northern blotting assays in this study. Firstly, we examined the expression of ALV-miRNA-p19-01 in DF-1 cells with or without ALV-J infection. As a result, ALV-miRNA-p19-01 was detected in cells which were infected with ALV-J, but not found in normal cells (Figure 2A). Subsequently, we detected ALV-miRNA-p19-01 using Northern blotting assay. As a result, the small RNAs corresponding to the pre-miRNA (50 nt) and the mature ALV-miRNA-p19-01 (23 nt) were clearly observed in DF-1 cells which were infected with ALV-J (Figure 2B). Next, we employed stem-loop RT-PCR to examine the expression of ALV-miRNA-p19-01 in various organs of cocks infected with ALV-J. Interestingly, ALV-miRNA-p19-01 was detected in the kidney, thymus, heart, spleen and bursa of Fabricius (Figure 2C). These data provided solid evidence for the existence of ALV-miRNA-p19-01 encoded by the genome of ALV-J.

As Drosha and Dicer are two key components in miRNA biogenesis, we determined whether ALV-miRNA-p19-01 was generated via the canonical miRNA biogenesis pathway. DF-1 cells with lower expression of Drosha and Dicer by siRNA were infected with ALV-J SCAU1903, and subjected to RT-PCR assay. The gga-miR-375 was used as a canonical control. Consistent with the canonical control, ALV-miRNA-p19-01 expression was significantly decreased when Drosha and Dicer expression were suppressed (Figure 2D). Furthermore, we used a universal inhibitor of RNA polymerase II, α-amanitin. DF-1 cells were infected with ALV-J SCAU1903 and subsequently treated with α-amanitin. RNAs were collected and analyzed by RT-qPCR. The results indicated that ALV-miRNA-p19-01 expression significantly decreased after treatment with α-amanitin (Figure 2E). These findings indicated that ALV-miRNA-p19-01 was processed by the canonical miRNA biogenesis pathway.

### 3.3. ALV-miRNA-p19-01 Promotes Viral Replication

As ALV-miRNA-p19-01 was encoded by the gag gene of ALV-J, we expanded our investigation to study whether ALV-miRNA-p19-01 involved viral replication. P27 is the structural protein of ALV-J, which was the most abundant protein in the virions and easier to detect. Therefore, we transfected DF-1 cells with ALV-miRNA-p19-01 mimics or controls, followed by the infection of ALV-J SCAU1903. As a result, compared to the controls, overexpression of ALV-miRNA-p19-01 effectively increased the expression of viral protein p27 and the transcriptional level of the viral genome (Figure 3A,B). Similar results were obtained using a TCID_50_ assay (Figure 3C). Furthermore, we investigated the effect of ALV-miRNA-p19-01 on ALV-J replication using electron microscopy (EM). DF-1 cells were transfected with ALV-miRNA-p19-01 mimic or ncRNA, followed by infection with ALV-J SCAU1903, and then processed for EM. As a result, compared to that of controls, virus-like particles were observed in cells with ALV-J infection, and the large-scale surface clustering of mature virions was observed in the ALV-miRNA-p19-01-expressing cells (Figure 3D), indicating that more virions were released to the medium in ALV-miRNA-p19-01-expressing cells.

To further study the function of ALV-miRNA-p19-01 during ALV-J infection, we constructed a replication-competent plasmid pVAX-ALV-SCAU1903-mut, with mutation of the precursor sequence of ALV-miRNA-p19-01 (Appendix A). DF-1 cells receiving pVAX-ALV-SCAU1903-mut DNA could produce the rescued virus rSCAU1903-mut. To validate the expression of ALV-miRNA-p19-01, we employed the Northern Blot Assay. As shown in Appendix A, ALV-miRNA-p19-01 was detected in DF-1 cells with infection of wild type virus ALV-J SCAU1903, but not mutated virus rSCAU1903-mut. Subsequently, we transfected DF-1 cells with ALV-miRNA-p19-01 mimics or controls, and followed the infection of rSCAU1903-mut. As a result, the virions in the culture supernatant were increased by ectopic expression of ALV-miRNA-p19-01 (Appendix A). Moreover, compared with that of controls, the expression of ALV-miRNA-p19-01 increased the number of virus particles in the culture supernatants (Appendix A). In addition, we explored the impact of ALV-miRNA-p19-01 on the replication of ALV-A, the results were similar to that of ALV-J (Appendix A). All these findings suggested that the ALV-miRNA-p19-01 played an important role in viral replication.

To further verify the effect of ALV-miRNA-p19-01 on ALV-J replication, we blocked ALV-miRNA-p19-01 using an ALV-miRNA-p19-01-specific antisense oligonucleotide (ASO) in DF-1 cells. As a result, the transfection of ASO-ALV-miRNA-p19-01 could efficiently reduce ALV-miRNA-p19-01 levels (Appendix A). Blocking ALV-miRNA-p19-01 markedly decreased the expression of viral protein p27 and the transcriptional level of viral particles from the supernatants (Figure 3E,F). Moreover, the subsequent release of progeny ALV-J particles to the cell supernatant was equally decreased by the transfection of ASO-ALV-miRNA-p19-01 which was measured by TCID_50_ assay (Figure 3G). We carried out the same work for ALV-J rSCAU1903-mut, which could not encode the ALV-miRNA-p19-01. The results showed there was no influence on ALV-J rSCAU1903-mut virion replication in the presence of ASO-ALV-miRNA-p19-01 (Appendix A). Together, these results demonstrated that ASO-ALV-miRNA-p19-01 did not affect the replication of ALV-J, but could effectively block the effect of ALV-miRNA-p19-01, which was expressed by ALV-J SCAU1903, and reduced the replication of ALV-J SCAU1903 virions.

### 3.4. ALV-miRNA-p19-01 Directly Targets DUSP6

Identifying the targets of ALV-miR-p19-01 is essential to understanding the function of the novel miRNA during ALV-J replication. The RNA-pull down and RNA-seq results showed that DUSP6 mRNA enrichment between two groups was different (Figure 4A). Bioinformatics analysis showed that the 3′UTR region of DUSP6 contains a site that is complementary with the seed sequences of ALV-miR-p19-01. Since miRNA usually serves as a guide RNA to target mRNAs to degrade mRNAs or inhibit translation, it was logical to examine the relationship between ALV-miR-p19-01 and DUSP6. Therefore, we constructed firefly luciferase reporter pmirGLO-DUSP6-3′UTR-WT and a mutant plasmid pmirGLO-DUSP6-3′UTR-MUT with mutations in the seed region. The luciferase activity analysis showed that ALV-miR-p19-01 markedly inhibited the luciferase activities of pmirGLO-DUSP6-3′UTR-WT, but did not affect the luciferase activities of pmirGLO-DUSP6-3′UTR-MUT (Figure 4B), indicating that DUSP6 was targeted by ALV-miR-p19-01. Furthermore, compared to that of controls, the overexpression of ALV-miR-p19-01 significantly reduced the expression of DUSP6 at mRNA and protein levels (Figure 4C,D). Taken together, these data indicated that DUSP6 was a direct cellular target of ALV-miR-p19-01 in DF-1 cells.

### 3.5. DUSP6/ERK2 Involves ALV-J Replication

It has been reported that ERK2 phosphatase activity is required for ALV replication [22,23]. To verify these results, we treated DF-1 cells with a selective inhibitor of ERK2 kinases MK-8353, followed by infection with ALV-J rSCAU1903-mut. As shown in Appendix A, the ERK2 phosphatase activity was inhibited by MK-8353, but not DMSO (Appendix A). Compared to that of controls, the viral protein P27 of virions in the culture supernatants was significantly decreased in the MK-8353-treated cells. Similar results were observed in the viral titers of culture supernatants. These data indicated that the inhibition of ERK2 phosphatase activity by MK-8353 affects viral replication.

DUSP6 has been reported to act as a negative regulator for ERK2 activity [24]. To investigate the effect of DUSP6 on ALV-J replication, we constructed an eukaryotic expression plasmid pRK5-flag-DUSP6, and transfected it into DF-1 cells, followed by infection with ALV-J rSCAU1903-mut. As a result, compared to that of controls, the overexpression of DUSP6 significantly decreased the phosphorylation of ERK2, and inhibited the secreted viral protein level (p27) in the culture supernatants (Figure 5A). Furthermore, both the transcriptional level of viral genome and the viral titers in the culture supernatant were markedly decreased by DUSP6 (Figure 5B,C). On the contrary, the knockdown of DUSP6 by si-RNA significantly enhanced the phosphorylation of ERK2 (Figure 5D). Moreover, the viral protein P27 of virions in the culture supernatants was significantly increased by the ectopic expression of si-RNA for DUSP6, but not si-NC (Figure 5D–F). Similar results were obtained using electron microscopy (Figure 5G). These results suggested that DUSP6 inhibits ALV-J replication by regulating ERK2 activity.

### 3.6. ALV-miR-p19-01 Promotes Viral Replication via Activating DUSP6/ERK2

To identify that the regulatory function of ALV-miR-p19-01 on viral replication was regulation of DUSP6 expression, we transfected ALV-miR-p19-01 mimics into DF-1 cells, followed by infection with ALV-J rSCAU1903-mut. Compared to that of controls, the upregulation of ALV-miR-p19-01 significantly decreased the expression of DUSP6, and increased the phosphorylation level of ERK2. Interestingly, the viral protein P27 of virions in the culture supernatants was significantly increased in the presence of ALV-miRNA-p19-01 (Figure 6A). However, the downregulation of ALV-miR-p19-01 by ASO-ALV-miRNA-p19-01 markedly increased the expression of DUSP6, decreased the phosphorylation of ERK2, and decreased the viral protein P27 of virions in the culture supernatants (Figure 6B). To further clarify that the role of DUSP6 in ALV-miR-p19-01 promoted viral replication, we transfected pRK5-flag-DUSP6 or pRK5-flag with ALV-miR-p19-01 mimics. As a result, the restitution of DUSP6 expression significantly decreased the phosphorylation level of ERK2 in DF-1 cells compared with the relative NC control, and markedly decreased the secreted viral protein level (p27) in the culture supernatants (Figure 6C). Clearly, restitution of DUSP6 expression significantly inhibited ALV-J replication compared with the relative NC control. Taken together, the results suggest that ALV-miR-p19-01 activates DUSP6/ERK2 by decreased DUSP6 and promotes ALV-J replication in DF-1 cells.

## 4. Discussion

As intracellular obligate parasites, viruses need to exploit the host cellular machinery for their replication. Virus interaction with host cell molecules is the key process in determining infection status during viral infection. In recent years, an increasing amount of evidence has revealed that virus-encoded miRNAs play key roles both in cellular and viral replication [25,26,27]. Whether RNA viruses produce functional miRNA-like molecules has been a controversial problem for several years, and the question remains debatable [27]. However, several recent reports using in silico analyses have suggested that RNA viruses are able to encode pre-miRNAs and mature miRNAs [28,29]. Additionally, miRNA-like small RNA from IAV and retroviruses have been functionally validated [30]. Thus, despite some ongoing controversy, evidence of the ability of RNA viruses to produce miRNA-like molecules is growing.

The retrovirus genome contains a single stranded RNA, encoding a reverse transcriptase and an integrase responsible for the insertion of the proviral DNA into the host genome [31,32]. Previous reports have indicated that retroviruses could encode viral miRNAs, such as HIV-1, Bovine Leukemia Virus (BLV) and Bovine Foamy Virus (BFV) [33,34,35]. HIV-1 nef miRNA from AIDS patients has been reported to possibly suppress both Nef function and HIV-1 virulence through the RNAi pathway [33]. In leukemic B cells, 10 miRNAs processed from a cluster of five hairpin structures in BLV had been found by deep sequencing and these miRNAs were associated with tumors [34]. Moreover, BFV could express three viral miRNAs (miR-BF1-3p, miR-BF1-5p and miR-BF2-3p) both in culture and in vivo. Deletion of the miRNA cassette in the U3 region of BFV reduced virus replication, which showed BFV-encoded miRNAs played an important role in virus replication [35]. As a member of the retroviruses, the ALV genome was confirmed to encode miRNA. A novel small RNA encoded by the E (XSR) element in ALV-J-transformed cell lines IAH30 (designated E (XSR) miRNA) was identified in 2014 [13]. In this study, ALV-miRNA-p19-01 was screened by deep sequencing from ALV-J-infected DF-1 cells, and confirmed by Northern blotting and stem-loop qRT-PCR. Interestingly, several engineered RNA viruses could encode miRNA-like RNAs [36,37,38,39]. More effort will be required to screen the RNA virus-encoded miRNAs.

MiRNAs can be found as isolated transcript units or clustered and co-transcribed as polycistronic primary transcripts, regulating the expression of target genes post-transcriptionally through the degradation of mRNAs or the inhibition of translation by binding to complementary sites in the 3′UTRs of mRNAs. Usually, miRNAs are transcribed by RNA polymerase II into a primary transcript. Subsequently, the pre-miRNA harpin is exported out of the nucleus and processed by the nuclear microprocessor complex. Interestingly, a non-canonical biogenesis pathway for miRs-B has been discovered. Pre-miRs-B could be transcribed by RNA polymerase III (RNAP III) from compact RNAP III type II genes. The 5′ short subgenomic transcripts of BLV are transcribed by RNA polymerase III to produce pre-miRNA, which is usually processed into mature miRNAs in a Drosha-independent manner [34]. However, the transcripts from the ALV genome were produced by RNA Pol II, and required Drosha and Dicer for further processing [13]. Our results also confirmed this phenomenon. More effort will be required to reveal the underlying mechanism in the future.

MiRNAs usually serve as guide RNAs to target mRNAs bearing complementary sequences. The 5′ end of a miRNA (“seed” region) plays a critical role in this biological progress. It is estimated that ∼60% of regulation by a particular miRNA is due to binding with perfect seed complementary to the target transcript [40]. Several pieces of evidence indicate that microRNAs play critical roles in regulating post-transcriptional translation by regulating their target genes. To explore the target genes of ALV-miRNA-p19-01, bioinformatic prediction, RNA-deep sequencing and luciferase reporter assay were employed in this study. DUSP6 was identified as a critical downstream target of ALV-miRNA-p19-01. Still, we did not obtain any solid evidence of the presence of other specific target genes. Based on results from prediction using bioinformatic prediction and RNA-deep sequencing, ChaC glutathione specific gamma-glutamylcyclotransferase 2 (CHAC2), histidine triad nucleotide binding protein 2 (HINT2), death associated protein kinase 2 (DAPK2), EBF transcription factor 1 (EBF1) and autophagy related 3 (ATG3) may act as the target genes of ALV-miRNA-p19-01, indicating that ALV-miRNA-p19-01 targets more genes than expected. More effort will be required to focus on the target genes of ALV-miRNA-p19-01 in the future.

The replication cycle of ALV-J can be divided into two phases: the early phase includes the events that occur from the virus binding to the surface of the host cell to the viral DNA integrating into the host cell genome. The late phase encompasses the events that occur from gene expression to release and maturation of new virions. The release step is mediated by cellular ESCRT machinery, which is hijacked by Gag [41]. PPXY, and LYPXnL, which are called late domains in retroviruses Gag, and serve as docking sites for the cellular ESCRT machinery, play the dominant role in ALV release [42]. In addition, there is accumulating evidence suggesting that the ubiquitination of ALV Gag is an important part of the release process, and the ubiquitination of Gag is dependent upon Nedd4 binding to the L domain [43]. Interestingly, it has been shown that the concentration of free ubiquitin in ALV particles is higher than that in the cytoplasm of the host cell, suggesting that ubiquitin is somehow recruited into virus particles [44]. We found that the large-scale surface clustering of mature virions 249 was observed in the ALV-miRNA-p19-01-expressing cells, which hinted that ALV-miRNA-p19-01 might affect viral release.

DUSP6 could act as a negative regulator of the extracellular regulated MAP kinase (ERK) signaling pathway [45], playing an important role in a variety of cancers, such as lung cancer and prostate cancer [46,47]. In prostate cancer, the degradation of DUSP6 played a key role in regulating the extension and duration of ERK activation, and promoted prostate tumorigenesis [48]. The ERK is an important effector of the Mitogen-Activated Protein Kinase (MAPK) signaling pathway [49]. The activation of ERK could affect cellular proliferation and apoptosis [50,51]. Furthermore, the ERK signaling pathway involves the life cycle of many viruses, including retrovirus, coronavirus, poxvirus, and coxsackievirus [52,53,54,55,56]. ERK-2 could promote the phosphorylation level of the matrix protein of human immunodeficiency virus type 1 (HTLV-1), and increase viral release and budding efficiency [57]. ERK-2 also regulated viral assembly and release by phosphorylating the p6 protein of HIV-1 [58,59]. Previous studies reported that ERK2/MAPK pathway involves ALV replication [22,23]. In this study, we confirmed this opinion.

In summary, a novel miRNA ALV-miRNA-p19-01 encoded by ALV-J SCAU1903 strain was identified, which promotes viral replication during ALV-J infection. Furthermore, ALV-miRNA-p19-01 promotes ALV-J replication in DF-1 cells by directly targeting DUSP6/ERK2. Overall, these results reveal the new mechanisms which are exploited by viruses to regulate the viral life cycle.

## Figures and Tables

**Figure 1 viruses-14-00805-f001:**
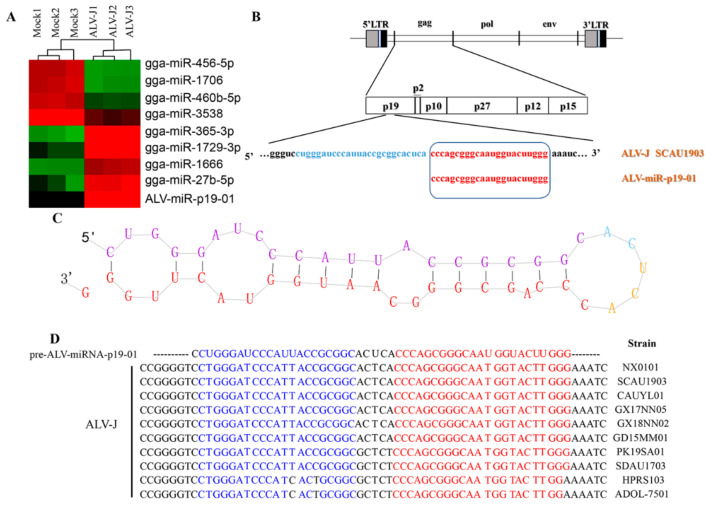
Identification of ALV-J-encoded miRNAs by deep sequencing. (**A**) Cluster analysis showing the differentially expressed miRNAs between ALV-J-infected DF-1 cells (ALV-J) and normal DF-1 cells (Mock). (**B**) Schematic diagram of the ALV-miRNA-p19-01 location on the ALV-J SCAU1903 genome. (**C**) Secondary-structure prediction of ALV-miRNA-p19-01. (**D**) Conserved analysis of ALV-miRNA-p19-01 among different ALV-J strains.

**Figure 2 viruses-14-00805-f002:**
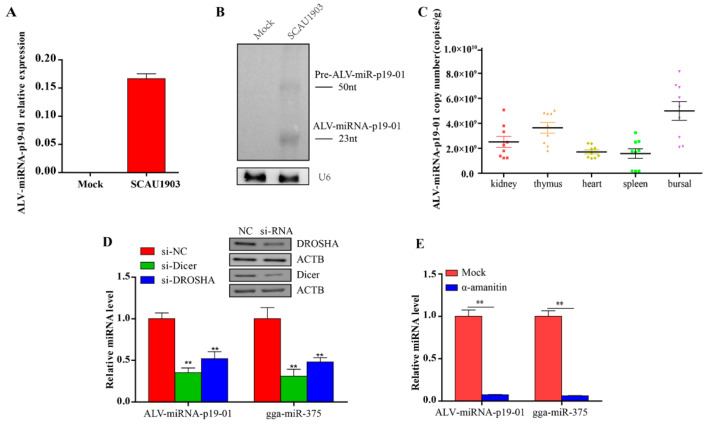
Experimental validation of ALV-miRNA-p19-01. (**A**,**B**) DF-1 cells were infected with ALV-J SCAU1903, normal cells as a negative control. (**A**) Mature ALV-miRNA-p19-01 were examined by stem-loop-RT-PCR in DF-1 cells. (**B**) ALV-miRNA-p19-01 was detected with Northern blotting. The probe was the biotin-conjugated antisense strand of ALV-miRNA-p19-01. Left: 30 μg of small RNA from normal DF-1 cells; Right: 30 μg of small RNA from DF-1 cells infected with ALV-J SCAU1903. U6 was used as a control. (**C**) The ALV-miRNA-p19-01 expression levels were measured by RT-PCR in different tissues of ALV-J-positive chickens. (**D**) Dicer and Drosha were suppressed with siRNAs in DF-1 cells, and ALV-miRNA-p19-01 expression was detected by RT-PCR. NC, normal control. (**E**) DF-1 cells were infected with ALV-J SCAU1903 and then treated with α-amanitin. ALV-miRNA-p19-01 expression was detected by RT-PCR. Statistically significant values are indicated by ** *p* < 0.01.

**Figure 3 viruses-14-00805-f003:**
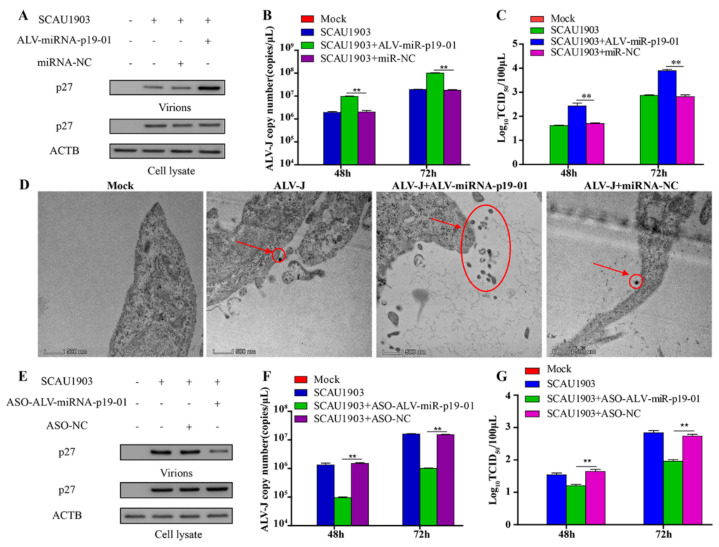
ALV-miRNA-p19-01 promotes ALV-J SCAU1903 replication. (**A**–**C**) DF-1 cells respectively transfected with ALV-miRNA-p19-01 mimic and miR-NC were infected with ALV-J SCAU1903 at 4 hpt. (**A**) The expression of p27 protein were examined by Western Blot. (**B**) The transcriptional level of viral particles in supernatants were measured by RT-PCR. (**C**) The supernatants from infected cells were titrated in DF-1 cells by ELISA for TCID50. (**D**) ALV-miRNA-p19-01 mimic was transfected into DF-1 cells and miR-NC was used as a control. DF-1 cells were infected with ALV-J SCAU1903 at 4 hpt and processed for transmission electron microscopy as described in Materials and Methods. (**E**–**G**) DF-1 cells respectively transfected with ASO-ALV-miRNA-p19-01 and ASO-NC were infected with ALV-J SCAU1903 at 4 hpt. (**E**) The expression of p27 protein were examined by Western Blot. (**F**) The transcriptional level of viral particles in supernatants were measured by RT-PCR. (**G**) The supernatants from infected cells were titrated in DF-1 cells by ELISA for TCID50. Statistically significant values are indicated by ** *p* < 0.01.

**Figure 4 viruses-14-00805-f004:**
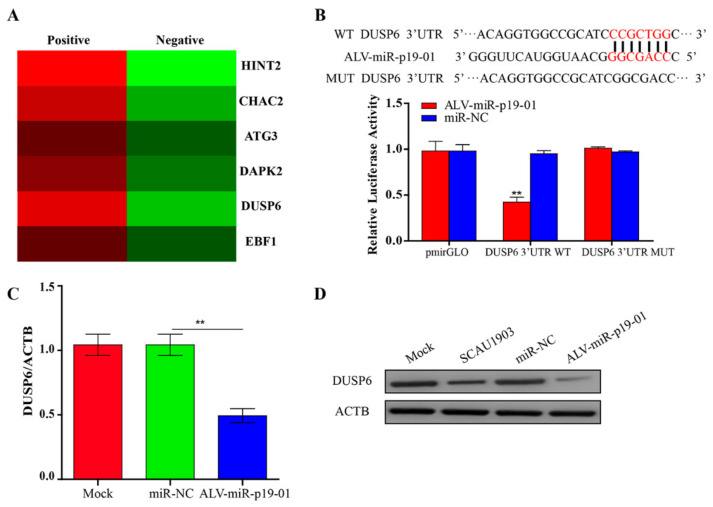
ALV-miRNA-p19-01 targets DUSP6 in DF-1 cells. (**A**) Putative candidate genes enriched from those ALV-miRNA-p19-01 binding genes obtained from RNA-pull down and RNA-sequence. (**B**) Luciferase reporter assays. Bioinformatics predictions of binding sites in DUSP6 3′UTR region. Wild-type (wt) and mutant-type (mut) sequences are indicated. DF-1 cells were co-transfected with wt or mut DUSP6 3′-UTR and ALV-miRNA-p19-01 mimic, nonspecific mimic (NC) was used as a control. The data are expressed as relative firefly luciferase activity normalized to Renilla luciferase activity. The data are presented as mean ± standard deviation values from 3 independent experiments. (**C**) DF-1 cells were transfected with ALV-miRNA-p19-01 mimic or miR-NC. The mRNA level of DUSP6 was measured by qRT-PCR. (**D**) DF-1 cells were transfected with ALV-miRNA-p19-01 mimic or miR-NC or infected with ALV-J SCAU1903. The protein level of DUSP6 was measured by Western Blot. β-actin was used as a control. Statistically significant values are indicated by ** *p* < 0.01.

**Figure 5 viruses-14-00805-f005:**
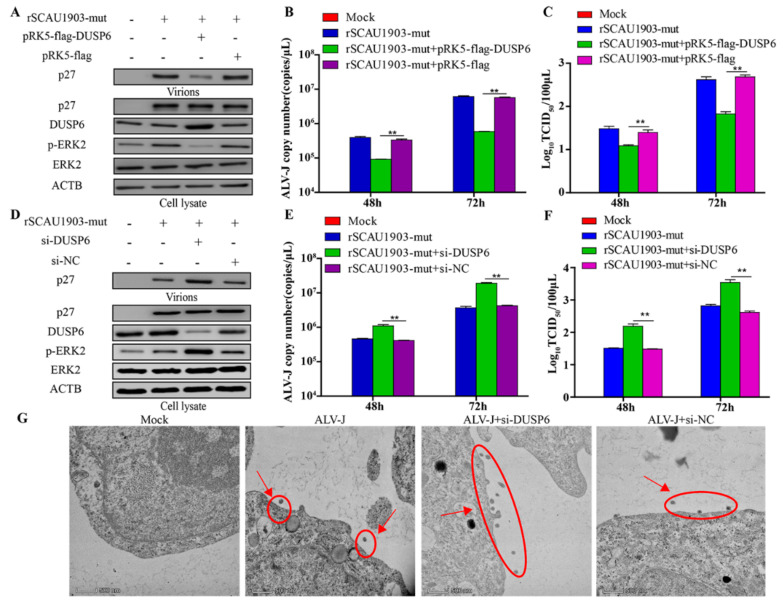
DUSP6/ERK2 pathway inhibits ALV-J replication. (**A**–**C**) DF-1 cells respectively transfected with pRK5-flag-DUSP6 plasmid and pRK5-flag plasmid were infected with ALV-J rSCAU1903-mut at 4 hpt. (**A**) The expression of p27 protein were examined by Western Blot. (**B**) The transcriptional level of viral particles in supernatants were measured by RT-PCR. (**C**) The supernatants from infected cells were titrated in DF-1 cells by ELISA for TCID_50_. (**D**–**F**) DF-1 cells respectively transfected with si-DUSP6 and si-NC were infected with ALV-J rSCAU1903-mut at 4 hpt. (**D**) The expression of p27 protein were examined by Western Blot. (**E**) The transcriptional level of viral particles in supernatants were measured by RT-PCR. (**F**) The supernatants from infected cells were titrated in DF-1 cells by ELISA for TCID_50_. (**G**) Si-DUSP6 was transfected into DF-1 cells and si-NC was used as a control. DF-1 cells were infected with ALV-J rSCAU1903-mut at 4 hpt and processed for transmission electron microscopy as described in Materials and Methods. Statistically significant values are indicated by ** *p* < 0.01.

**Figure 6 viruses-14-00805-f006:**
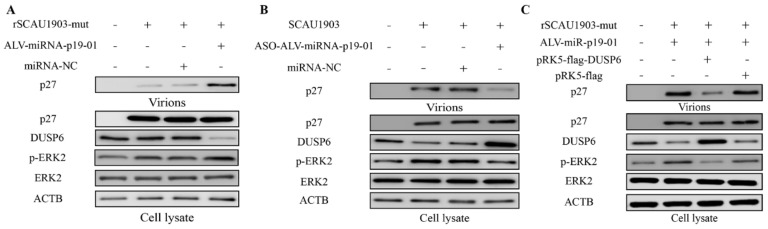
ALV-miR-p19-01 promotes viral replication via activating DUSP6/ERK2 pathway. (**A**) DF-1 cells respectively transfected with ALV-miRNA-p19-01 mimic and miR-NC were infected with ALV-J rSCAU1903-mut at 4 hpt. (**B**) DF-1 cells respectively transfected with ASO-ALV-miRNA-p19-01 mimic and ASO-NC were infected with ALV-J SCAU1903 at 4 hpt. (**C**) DF-1 cells were co-transfected with ALV-miRNA-p19-01 mimic and pRK5-flag-DUSP6, and then DF-1 cells were infected with ALV-J rSCAU1903-mut at 4 hpt. The expressions of p27, DUSP6, ERK2, p-ERK2 and β-actin were detected by Western Blot.

## Data Availability

All data generated or analyzed during this study are included in this published article and its Appendix A. The RNA-seq data can be found in NCBI (accession SRR10762446, SRR10762447 and GSE186119).

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
