# Peer review of "ALV-miRNA-p19-01 Promotes Viral Replication via Targeting Dual Specificity Phosphatase 6"

_viruses, 2022, doi:10.3390/v14040805_

Round 1

Reviewer 1 Report

This is an interesting and novel paper that appears to be well supported

English editing is needed.  Multiple inappropriate expressions noted, including:

  1. line 30:  fit into host genome (integrated)
  2.  line 39:  multifarious (multiple)
  3.  line 40:  prooved (prooven)
  4. line 42:  Till (until)

Author Response

Point 1: line 30:  fit into host genome (integrated)

Response 1: Thank you for the comment. The precedent version of the sentence has been replaced, becoming “The genome of retroviruses usually integrates with the host genome”.

Point 2: line 39:  multifarious (multiple)

Response 2: Thank you for the comment. We have revised it as suggested.

Point 3: line 40:  proved (proven)

Response 3: Thank you for the comment. We have revised it as suggested.

Point 4: line 42:  Till (until)

Response 4: Thank you for the comment. The precedent version of the sentence has been replaced, becoming “To data, more than 530 virus-encoded miRNAs have been identified”.

Reviewer 2 Report

In their manuscript “ALV-miRNA-p19-01 promotes viral release via targeting dual specificity phosphatase 6“, Yan et al. described the identification of a new miRNA (designed ALV-miRNA-p19-01) encoded by avian leukosis virus J (ALV-J). This miRNA was processed by the canonical pathway and promoted replication of the ALV-J in vitro. As a direct target of ALV-miRNA-p19-01, authors suggested the chicken dual specificity phosphatase 6 (DUSP6), siRNA knock-down of which simulated the effect of ALV-miRNA-p19-01. The down-regulatory effect of DUSP6 on ALV-J replication was accompanied by a decreased phosphorylation of ERK2, which authors propose as a mechanism behind the ALV-miRNA-p19-01 action. Overall, the main findings of the study are very interesting and well supported by experiments. On the other hand, I have some concerns to be solved before acceptance of the manuscript por publication in Viruses.

  1. I cannot see any weighty reason for analyzing the presence of miRNAs in exosomes prepared from cockerel seminal fluid. The study should logically start with miRNA analysis in ALV-J-infected DF-1 cells and tissues of ALV-J-positive chickens. The chapter 3.1 could be simply deleted together with the respective parts of methods (2.4) and figures (S1).
  2. The identification of DUSP6 as a direct target of ALV-miRNA-p19-01 using the prediction software, RNA sequence analysis and RNA pull-down is not described either in results (3.5) or Figure 4. Description of the RNA pull-down is missing in methods.
  3. Without a thorough analysis, authors cannot insist that the ALV-miRNA-p19-01 acted at the late phase of the virus replication cycle and promoted virus release. This could be discussed, but not given as a conclusion supported by experiments.
  4. The authors documented that ALV-miRNA-p19-01 was processed canonically by Dicer and Drosha RNases. In addition to these experiments, authors should easily test the transcription by either RNA polymerase II or III with Pol II inhibition by α-amanitin.
  5. All DNA alignments showing the conservation of ALV-miRNA.p19-01 sequence should contain the prototype strain of ALV-J (HPRS103) and the American ADOL strains. Showing only the derived strains of Asian origin limits the impact of the study. In figures (1D, S2), the names of strains instead of the accession numbers should be given.
  6. Fig. S2 shows that the sequence of ALV-miRNA-p19-01 is conserved also in other ALV subgroups (B, E, and K) whereas mutated in ALV-A subgroup. It would be nice to test if ALV-miRNA-p19-01 promotes replication of other ALV subgroups. What about the ALV-C subgroup? Fig. S2 should be discussed (and referred to) in the text.

Minor concerns

In Figure 1D, the passenger strand of the pre-miRNA should be highlighted together with the miRNA strand in a conventional way.

p27 was analyzed both in cell lysates and in virions. The importance of this should be mentioned for a benefit of readers. The preparation of virions for Western blot should be described in 2.10.

The TEM images in Figures 3D and 5G are too small.

The Discussion part of the manuscript is too extensive and contains many unnecessary details. On the other hand, better comparison with the study Yao et al. 2014 (Ref. 13) would be useful.

The language should be improved in terms of style and word usage.

Author Response

Point 1: I cannot see any weighty reason for analyzing the presence of miRNAs in exosomes prepared from cockerel seminal fluid. The study should logically start with miRNA analysis in ALV-J-infected DF-1 cells and tissues of ALV-J-positive chickens. The chapter 3.1 could be simply deleted together with the respective parts of methods (2.4) and figures (S1).

Response 1: Thank you for the comment. In fact, we firstly analyzed the presence of miRNAs in exosomes was to explore the differential miRNA expression between ALV-J-infected chickens and healthy chickens’ seminal fluid exosomes, and the article had been published in 2020 which named “MicroRNA expression profile in extracellular vesicles derived from ALV-J infected chicken semen”. In this work, we re-analyzed the data mentioned above and found a novel miRNA in ALV-J positive seminal fluid exosomes.

Point 2: 2.The identification of DUSP6 as a direct target of ALV-miRNA-p19-01 using the prediction software, RNA sequence analysis and RNA pull-down is not described either in results (3.5) or Figure 4. Description of the RNA pull-down is missing in methods.

Response 2: Thank you for underlining this deficiency. We have rewritten the Materials and Methods section. The description of the RNA pull-down was added in line 153-158.

Point 3: Without a thorough analysis, authors cannot insist that the ALV-miRNA-p19-01 acted at the late phase of the virus replication cycle and promoted virus release. This could be discussed, but not given as a conclusion supported by experiments.

Response 3: Thank you for the comment. This section was revised and modified according to the information showed in the work suggested by the reviewer. The title has been replaced, becoming” ALV-miRNA-p19-01 promotes viral replication via targeting dual specificity phosphatase 6”.

Point 4: The authors documented that ALV-miRNA-p19-01 was processed canonically by Dicer and Drosha RNases. In addition to these experiments, authors should easily test the transcription by either RNA polymerase II or III with Pol II inhibition by α-amanitin.

Response 4: Thank you for the comment. We have explored the impact of α-amanitin on the transcription of ALV-miRNA-p19-01. The results were showed in Fig.3F.

Point 5: All DNA alignments showing the conservation of ALV-miRNA.p19-01 sequence should contain the prototype strain of ALV-J (HPRS103) and the American ADOL strains. Showing only the derived strains of Asian origin limits the impact of the study. In figures (1D, S2), the names of strains instead of the accession numbers should be given.

Response 5: Thank you for the comment. We have revised it as suggested.

Point 6: Fig. S2 shows that the sequence of ALV-miRNA-p19-01 is conserved also in other ALV subgroups (B, E, and K) whereas mutated in ALV-A subgroup. It would be nice to test if ALV-miRNA-p19-01 promotes replication of other ALV subgroups. What about the ALV-C subgroup? Fig. S2 should be discussed (and referred to) in the text.

Response 6: Thank you for the comment. In fact, we had explored the function of ALV-miRNA-p19-01 on other ALV subgroups, including ALV-A, ALV-B and ALV-K and this part will be showed in another article; We have re-analyzed the sequence of ALV-miRNA-p19-01 in ALV-C and ALV-D, the results were showed in Fig.S2.

Minor concerns

Point 7: In Figure 1D, the passenger strand of the pre-miRNA should be highlighted together with the miRNA strand in a conventional way.

Response 7: Thank you for the comment. We have highlighted the passenger strand of the pre-miRNA in Figure 1D.

Point 8: p27 was analyzed both in cell lysates and in virions. The importance of this should be mentioned for a benefit of readers. The preparation of virions for Western blot should be described in 2.10.

Response 8: Thank you for the comment. We have added the information required as explained above:”P27 is the structural protein of ALV-J, which was the most abundant protein in virions and easier to detect” in line 257-259.

Point 9: p27 was analyzed both in cell lysates and in virions. The importance of this should be mentioned for a benefit of readers. The preparation of virions for Western blot should be described in 2.10.

Response 9: Thank you for the comment. We have added the information required as explained above:”P27 is the structural protein of ALV-J, which was the most abundant protein in virions and easier to detect” in line 257-259.

Point 10: The TEM images in Figures 3D and 5G are too small.

Response 10: Thank you for the comment. We have enlarged the TEM images in Figures 3D and 5G.

Point 11: The Discussion part of the manuscript is too extensive and contains many unnecessary details. On the other hand, better comparison with the study Yao et al. 2014 (Ref. 13) would be useful.

Response 11: Thank you for the comment. We have revised it as suggested.

Point 12: The language should be improved in terms of style and word usage.

Response 12: Thank you for the comment. The manuscript has been submitted to MDPI for English editing and the certificate was showed in the Supplementary.

Round 2

Reviewer 2 Report

Authors of the manuscript “ALV-miRNA-p19-01 promotes viral replication via targeting dual specificity phosphatase 6” substantially improved the text and answered most of my questions. Two my concerns, however, remain to be solved.

First (point 1 of my first review), authors did not explain why they started their analysis by exosomes prepared from cockerel seminal fluids and not directly DF-1 cells. In the rebuttal letter, authors declared that this was because of the previous study on miRNAs in exosomes. This, however, is not clear to the readers and complicates understanding of the findings. I still believe that the chapter 3.1 should be deleted and the study should start with miRNA analysis in ALV-J-infected DF-1 cells and tissues of ALV-J-positive chickens.

Second (point 6 of my first review), it remains to be shown if ALV-miRNA-p19-01 promotes replication of ALV subgroup A, where the respective sequence is mutated. Authors declare that they analyzed ALV-A subgroup, but plan to publish this separately. Because the ALV-A data might either confirm or refute the hypothesis on dual specificity phosphatase 6 effect, I strongly recommend that ALV-A data are shown in this manuscript.   

Author Response

Point 1: First (point 1 of my first review), authors did not explain why they started their analysis by exosomes prepared from cockerel seminal fluids and not directly DF-1 cells. In the rebuttal letter, authors declared that this was because of the previous study on miRNAs in exosomes. This, however, is not clear to the readers and complicates understanding of the findings. I still believe that the chapter 3.1 should be deleted and the study should start with miRNA analysis in ALV-J-infected DF-1 cells and tissues of ALV-J-positive chickens.

Response 1: Thank you for the comment. We have revised it as suggested.

Point 2: Second (point 6 of my first review), it remains to be shown if ALV-miRNA-p19-01 promotes replication of ALV subgroup A, where the respective sequence is mutated. Authors declare that they analyzed ALV-A subgroup, but plan to publish this separately. Because the ALV-A data might either confirm or refute the hypothesis on dual specificity phosphatase 6 effect, I strongly recommend that ALV-A data are shown in this manuscript.

Response 2: Thank you for the comment. We have explored the impact of ALV-miRNA-p19-01 on the replication of ALV-A, the results were showed in Fig.S3. We found that ALV-miRNA-p19-01 also could promotes replication of ALV-A.

This manuscript is a resubmission of an earlier submission. The following is a list of the peer review reports and author responses from that submission.